# Quality of life in elderly ICU survivors before the COVID-19 pandemic: a systematic review and meta-analysis of cohort studies

Kevin Ariyo ,[1] Sergio Canestrini,[2] Anthony S David,[3] Alex Ruck Keene,[1,4] Sebastian Wolfrum,[5,6] Gareth Owen[1]

For numbered affiliations see end of article.

**Correspondence to**
Kevin Ariyo;
kevin.ariyo@kcl.ac.uk

## ABSTRACT

**Objectives** The influence of age on intensive care unit (ICU) decision-making is complex, and it is unclear if it is based on expected subjective or objective patient outcomes. To address recent concerns over age-based ICU decision-making, we explored patient-assessed quality of life (QoL) in ICU survivors before the COVID-19 pandemic.

**Design** A systematic review and meta-analysis of cohort studies published between January 2000 and April 2020, of elderly patients admitted to ICUs.

**Primary and secondary outcome measures** We extracted data on self-reported QoL (EQ-5D composite score), demographic and clinical variables. Using a random-effect meta-analysis, we then compared QoL scores at follow-up to scores either before admission, age-matched population controls or younger ICU survivors. We conducted sensitivity analyses to study heterogeneity and bias and a qualitative synthesis of subscores.

**Results** We identified 2536 studies and included 22 for qualitative synthesis and 18 for meta-analysis (n=2326 elderly survivors). Elderly survivors' QoL was significantly worse than younger ICU survivors, with a small-to-medium effect size (d=0.35 (−0.53 and −0.16)). Elderly survivors' QoL was also significantly greater when measured slightly before ICU, compared with follow-up, with a small effect size (d=0.26 (−0.44 and −0.08)). Finally, their QoL was also marginally significantly worse than age-matched community controls, also with a small effect size (d=0.21 (−0.43 and 0.00)). Mortality rates and length of follow-up partly explained heterogeneity. Reductions in QoL seemed primarily due to physical health, rather than mental health items.

**Conclusions** The results suggest that the proportionality of age as a determinant of ICU resource allocation should be kept under close review and that subjective QoL outcomes should inform person-centred decision -aking in elderly ICU patients.

**PROSPERO registration number** CRD42020181181.

## INTRODUCTION

The influence that age should have on intensive care decision-making has been debated across policy and clinical practice.[1,2] Age associates (inversely) with the probability of intensive care unit (ICU) survival and length

**Strengths and limitations of this study**

► To our knowledge, this is the first systematic review and meta-analysis to explore quality of life (QoL) outcomes in elderly intensive care unit survivors and to explore sources of variation between these studies.

► To ensure consistency and policy relevance, we only included one type of measure within the meta-analysis (EQ-5D).

► With our large sample, we could estimate the population QoL with reasonable precision, as evidenced by narrow CIs.

► Wide prediction intervals suggest that our results should not be used to make individual-level predictions. Our sample had a mixture of conditions, and because data were reported inconsistently and often at study level, it is difficult to generalise to specific clinical groups, including patients with COVID-19.

of life after ICU,[3,4] outcomes generally considered to be relevant to resource allocation.[2] However, age is also a protected characteristic in several jurisdictions, and in England and Wales, resource allocation based on age must be a 'proportionate means of achieving a legitimate aim', if it is not to be contrary to the Equality Act (2010).

For elderly patients for whom admission to ICU is clinically appropriate, an important part of person-centred decision-making is for them, or their families, to be given information about the likely outcome of admission. Patients may seek to integrate survival and biomedical outcomes with subjective outcomes, including quality of life (QoL). Subjective QoL in elderly ICU survivors has been studied less frequently than these objective measures.[3,5] This is notable given that subjective QoL (via quality-adjusted life years or QALYs) is very influential in clinical resource allocation (eg, at the National Institute for Health and Care Excellence; NICE).

Person-centred decision-making requires consideration of patient experience since physician-rated QoL is not always well correlated with patient-rated QoL.

We considered a rapid review to be urgent because age is a strong risk factor for severe COVID-19 infection,[6] and severe COVID-19 has placed considerable pressure on ICU resource allocation[7] and is likely to do so in the future. Additionally, some have expressed concerns that elderly adults may have been disproportionately less likely to receive ICU before the pandemic.[1 2 8–10] As health system collapse remains a possibility, this raises the prospect of difficult triage decisions. In particular, services will need to weigh up various ethical positions to decide how important age is to these admission policies.[11] It is therefore important that older persons' subjective outcomes are better understood.

We conducted a meta-analysis on patient-reported QoL in elderly adults undergoing ICU. Following a systematic review, we addressed three questions:

1. At follow-up, do elderly ICU survivors have better/worse QoL compared with their scores before ICU?
2. At follow-up, do elderly ICU survivors have better/worse QoL than age-matched community controls?
3. At follow-up, do elderly ICU survivors have better/worse QoL than ICU survivors aged under 65?

Determining the effect of illness and ICU on QoL is complicated because QoL is itself influenced by many variables[12] and some are non-clinical. These influences are too complex to resolve completely, but where possible, we sought to model relevant variables (illness severity, ICU length of stay and mortality rate) as predictors of QoL in elderly ICU survivors at follow-up, compared with controls.

## METHODS
### Search strategy
We searched for English-language journal articles, published between January 2000 and April 2020. Six online bibliographical databases were used: Central, CINAHL, Cochrane Library, EMBASE, MEDLINE and PsycINFO. Our search followed a prepublished PROSPERO protocol.

The search terms focused on intensive care, elderly adults and QoL (see item 6 of the online supplemental appendix). We supplemented this with a forward citation and reference list search based on the eligible articles as well as consultation with experts.

### Patient and public involvement
No patient or public advisers were involved in this project.

### Selection criteria
We undertook study selection using EndNote X9 using a standardised crib sheet. See figure 1 for an overview. The inclusion and exclusion criteria are detailed further in item 6 of the online supplemental appendix.

At the title and abstract level, we identified potentially eligible studies that took place in an ICU and referred to either QoL or elderly adults. Full texts were eligible if (a) all participants underwent ICU; (b) there were at least 20 elderly patients and controls; (c) scores from a validated QoL scale were reported, for a group aged at least 60+, with at least 3-month follow-up review; (d) the follow-up QoL scores were derived from the patient, rather than a professional; and (e) the study reported QoL scores from the same scale for either the same patients before the ICU admission, age-matched community controls or ICU survivors aged under 65.

Where we could not include potentially eligible studies, due to poor reporting, we contacted study authors for unpublished data. We also considered whether to include studies that focused only on cardio-surgical or neurosurgical patients, given the effects of the diagnostic heterogeneity that characterises the reference population of the studies included in our review (general ICU patients with various conditions). However, none of these studies met the other inclusion criteria.

KA led the study selection at all stages, and a postdoctoral research assistant conducted reliability checks for 50% of full-text articles. We found nearly perfect inter-rater agreement, as measured by Cohen's kappa (k=0.86).[13] Queries were resolved by GO.

### Data extraction
One reviewer (KA) extracted relevant data from all eligible studies, recording this on a standardised spreadsheet. MK independently extracted data from 10% of eligible studies, to evaluate consistency. The primary outcome was the QoL composite scores. Secondary variables included demographics, QoL subscale scores, mortality (from ICU to follow-up), illness severity (either the Acute Physiology and Chronic Health Evaluation, APACHE-II; or the Simplified Acute Physiology Score, SAPS-II), length of ICU stay, length of hospital stay, and average follow-up time. When one dataset was used for multiple studies, we included the study with the clearest data reporting.

To ensure consistency, we included only composite scores from the EuroQoL health-related QoL instrument (EQ-5D) within the meta-analysis. Where possible, we also converted the eight subscales of the 36-item Short Form Survey (SF-36) to an EQ-5D Index Score, using an established mapping algorithm.[14] The remaining studies were included within the qualitative synthesis only.

### Data analysis
We explored the effect of age on EQ-5D composite scores using random-effect meta-analyses. KA conducted the analysis using R Statistics. We used the restricted maximum likelihood method to calculate the effect sizes (Cohen's d), which were weighted by the inverse of the sampling variance: meaning that studies with higher variance contributed less to the summary effect size. We interpreted these effect sizes using conventional criteria as a guide (0.2=small; 0.5=medium; 0.8=large).[15] We then

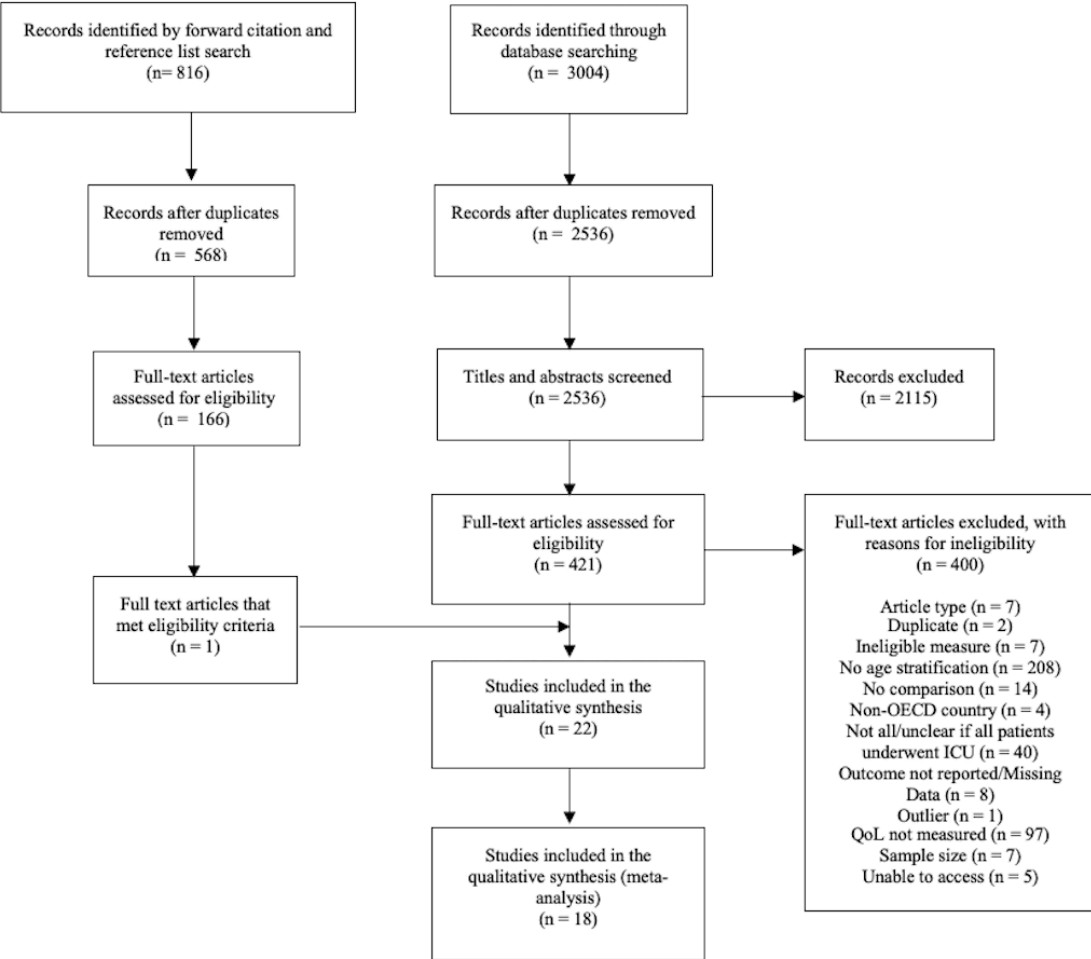

From: Moher D, Liberati A, Tetzlaff J, Altman DG, The PRISMA Group (2009). Preferred Reporting Items for Systematic Reviews and Meta-

**Figure 1** A Preferred Reporting Items for Systematic Reviews and Meta-Analyses flow diagram that outlines the study selection process. ICU, intensive care unit; OECD, Organisation for Economic Co-Operation and Development; QoL, quality of life.

conducted sensitivity analyses for each meta-analysis to assess risk of bias at the study level, including heterogeneity (eg, $I^2$ statistic), influential studies (eg, Cook's distance) and publication bias (funnel plots and Egger's test).

To investigate the remaining heterogeneity, we then conducted two secondary analyses: a moderator analysis to explore variation within a specific predictor and a random-effect meta-regression to explore relationships between multiple predictors.

We used several strategies to handle missing data. When the study only reported median values and IQRs, we estimated the mean and SD using conventional formulae.[16 17] When neither the SD nor IQR was reported, we estimated the SD using prognostic imputation.[18] This calculates the average of observed variances to estimate the missing SD

values. We excluded studies with missing data if these methods were inapplicable.

One reviewer (KA) assessed the methodological rigour of the included studies using an 11-item quality checklist (three irrelevant items were excluded).[19] The criteria were scored as either 2 (complete fulfilment), 1 (partial fulfilment) or 0 (not fulfilled). We then calculated a total score for each study and rated them as either high quality (17/22 or higher), moderate quality (between 10/22 and 16/22) or low quality (9/22 or lower). Queries were resolved through discussion with GO and SC.

For the qualitative synthesis, we defined a set of criteria for each measure to allocate subscores to either 'mental health' or 'physical health' categories. We then calculated a crude average for subscales within these two categories and weighted them on a scale of 1–100 (0=minimum QoL;

100=maximum QoL). As this approach is subjective, we present these findings only as a qualitative supplement.

This study follows methodological guidance from Preferred Reporting Items for Systematic Reviews and Meta-Analyses (see online supplemental appendix).

## RESULTS
### Descriptive statistics
After screening duplicates, the database search revealed 2536 records for title and abstract screening. From these, we reviewed 421 potentially relevant full-text articles for eligibility. Sixteen of these studies met the full criteria and were included in the initial meta-analysis. A further two studies were deemed eligible following a forward citation search and contact with study authors. This led to a total of 18 studies included in the initial meta-analysis (n=2326 elderly adults). Eleven of these studies reported age characteristics for the elderly patients (mean=79.04), while the others reported the minimum age only.

Most of the studies included both medical and surgical ICU patients (15 studies). The remaining studies focused on surgical (two studies) or medical (one study) patients only. A full breakdown of reasons for admissions is available in the online supplemental appendix.

Three types of outcome were included in the meta-analysis. These results compared QoL at follow-up to either pre-ICU scores (five studies), age-matched community controls (ten studies) or younger survivors of ICU (six studies). We provide a full summary in table 1.

For the qualitative analysis, we identified four further studies. Five different measurement scales were reported: the EuroQoL EQ-5D health-related QoL instrument (EQ-5D Utility Index or Visual Analogue Scale, eleven studies), the short-form medical outcome questionnaire (SF-36, eight studies), the Nottingham Health Profile (one study), the QoL Index (one study) and the WHO QoL instruments (WHOQOL-BREF, one study). SF-36 scores were converted to EQ-5D Index scores for the meta-analysis, while the other measures were excluded (see 'Methods' section).

### Meta-analyses
Table 2 outlines the results of the three meta-analyses. There was a significant difference in EQ-5D composite scores between elderly patients before and after ICU, with a small effect size (d=−26, p=0.005). This suggests that elderly patients may expect a slightly worse QoL at follow-up, compared with their own scores 1 month before ICU.

There was a marginally significant difference in EQ-5D composite scores between elderly ICU survivors and age-matched community controls, with a small effect size (d=−0.22, p=0.05). These results suggest that QoL may be slightly lower in elderly ICU survivors, relative to community controls.

Elderly ICU survivors (aged over 65) had significantly lower composite scores on the EQ-5D, compared with younger ICU survivors (aged under 65), with a small-to-medium effect size (d=−0.33, p<0.01). This suggests that on average, QoL in elderly ICU survivors is slightly worse than younger ICU survivors.

### Sensitivity analyses
We reviewed the impact of influential cases within each analysis. One study was excluded from the community meta-analysis as a substantial outlier and influential result. If the result had not been excluded, the effect size would have been stronger (d=−1.97—ie, a larger difference in QoL favouring younger controls) but non-significant (p=0.27), mainly due to large heterogeneity ($I^2$=100%). It is unclear why this study reported substantially outlying results, although the reported SDs were considerably lower than other studies.

After excluding this, one other study was somewhat influential within the community analysis (see online supplemental appendix). This study was retained as we acquired the full dataset, and we can therefore be confident of its reporting accuracy. If this study was excluded, the effect size would have been weaker (d=−0.13) and non-significant (0.010) in the same direction.

We identified no further outliers according to our criteria.

### Secondary analyses
There was moderate-to-large heterogeneity between studies. For significant results, we explored the role of other variables using post hoc subgroup analyses and meta-regressions. These results should be interpreted with caution, due to low sample sizes.

Length of follow-up significantly predicted greater differences in QoL between elderly ICU survivors and patients aged under 65 (k=6, p<0.001). This suggests that elderly survivors may have worse QoL in the long term and comparable QoL in the medium term.

The minimum age of the sample significantly predicted greater differences in QoL between elderly ICU survivors and age-matched community controls (k=10, p=0.02). Subgroup analyses revealed that in studies with only very old patients (aged 75–80+), elderly ICU survivors' QoL was no worse than their age-matched community controls (k=6, d=−0.06, p>0.05). In contrast, when elderly was defined as 65–70+, elderly ICU survivors had much worse QoL than age-matched community controls (k=4, d=0.45, p<0.03). This suggests that some of the variation was due to age differences in QoL in community controls.

Controlling for these variables reduced heterogeneity between studies by 10% and 47%, in both cases. No model significantly accounted for variance when the outlier was included in the community analysis.

Neither severity of illness, year of publication nor sex significantly accounted for heterogeneity between the studies, either individually or within a meta-regression (p>0.05).

**Table 1** The main characteristics of the studies and the relevant data included in the meta-analyses

| First author | Year | Country | N | Min age | % Male | Follow-up (avg. months) | ICU LoS (days) | Mortality | Severity (scaled avg.) | Raw measure | Comparison | Quality |
|---|---|---|---|---|---|---|---|---|---|---|---|---|
| Abelha | 2007 | Portugal | 112 | 65+ | 61.00% | 6 | | 28.00% | | SF-36 * | ICU survivors younger than 65 years old | M |
| Ali | 2018 | Australia | 32 | 65+ | 80.00%† | 12 | 5 | | 0.24 | EQ-5D | Age-matched South Australian controls | H |
| Andersen | 2015 | Norway | 53 | 80+ | 69.00% | 40.8 | 1.9 | 81.52% | 0.27 | EQ-5D | Age-matched and sex-matched Norwegian population | M |
| De Rooij | 2008 | Netherlands | 187 | 80+ | 51.00% | 44.4 | 1.29 | 61.52% | 0.21 | EQ-5D | Age-matched British population | M |
| Eddleston | 2000 | UK | 39 | 65+ | 52.45%† | 3 | | | | SF-36* | ICU survivors younger than 65 years old | M |
| Ferrao | 2015 | Portugal | 290 | 66+‡ | 26.00% | 27.6 | | | | EQ-5D | ICU survivors younger than 65 years old | M |
| Grace | 2007 | Australia/NZ | 99 | 60+ | NR | 28 | | 60.00% | 0.28 | EQ-5D | Retrospective patient ratings for 1 week before ICU | L |
| Hofhuis | 2011 | Netherlands | 49 | 80+‡ | 46.90% | 6 | 5.35 | 40.83% | 0.25 | SF-36* | Age-matched Dutch population and retrospective proxy ratings for 4weeks before ICU | M |
| Honselmann§ | 2015 | Germany | 352 | 65+ | 53.40% | 12 | 2.58 | 43.36% | | EQ-5D | ICU survivors younger than 65 years old | M |
| Honselmann[c-d] | 2015 | Germany | 291 | 65+ | 53.61% | 12 | 2.34 | 43.36% | | EQ-5D | Age-matched German controls | M |
| Jeitziner | 2015 | Switzerland | 124 | 65+ | 73.00% | 12 | 4.57 | | 0.29 | SF-36* | Age matched Swiss controls and retrospective patient ratings for 1 week before ICU | M |
| Kaarola | 2006 | Finland | 299 | 65+ | 75.00% | 47 | | 57.00% | | EQ-5D | ICU survivors younger than 65 years old | M |
| Levinson | 2016 | Australia | 322 | 80+ | 58.00%† | 24 | 1.28 | 21.45% | | SF-36* | Age-matched and sex-matched Australian population | H |
| Merlani | 2007 | Switzerland | 36 | 70+ | 52.00% | 24 | 3.00 | 63.00% | 0.26 | EQ-5D | Age-matched Swiss population | M |

**Table 1** Continued

| First author | Year | Country | N | Min age | % Male | Follow-up (avg. months) | ICU LoS (days) | Mortality | Severity (scaled avg.) | Raw measure | Comparison | Quality |
|---|---|---|---|---|---|---|---|---|---|---|---|---|
| Oeyen | 2007 | Netherlands | 63 | 80+ | 60.00%† | 12 | | 49.60% | 0.26 | EQ-5D | Retrospective patient or proxy ratings for 1 week before ICU | M |
| Sacanella | 2011 | Spain | 112 | 65+ | 57.00% | 12 | 3.35 | 48.70% | 0.27 | EQ-5D | Retrospective patient or proxy ratings before feeling ill and requiring ICU | M |
| Schroder | 2011 | Denmark | 36 | 75+ | 56.00% | 12 | 9.4 | 53.85% | | SF-36* | Age-matched Danish population | L |
| Sznajer | 2001 | France | 65 | 65+‡ | 55.90%† | 6 | | | | EQ-5D | ICU survivors younger than 65 years old | M |
| Villa | 2016 | Spain | 54 | 75+ | 50.00% | 12 | | 43.18% | 0.23 | SF-36* | Spanish population aged 75+ | M |
| *Weighted avg.* | | | 128.53 | 69.50 | 55.74% | 22.98 | 3.02 | 44.92% | 0.26 | | | |
| *Range* | | | 23–352 | 60–80 | 26%–80% | 3–100.8 | 1.28–9.4 | 21.45%–81.52% | 0.12–0.34 | | | |

See above for measures. Unless specified, we do not report data where it is not representative of at least 66.67% of the included sample.
Where the manuscript did not report the relevant information, we marked this as NR.
*Converted to EQ-5D composite score.
†Reported for study level only so not included in meta-analysis.
‡Combined elderly groups.
§We analysed some unpublished data from Honselmann *et al*; therefore, we have presented descriptions for the full dataset only, without quality assessment.
¶In the Honselmann study, the sample for the community study was slightly smaller than for the young/old comparison.
H, high quality; ICU, intensive care unit; L, low quality; LoS, length of stay; M, moderate quality.

**Table 2** A summary of effect sizes, CIs, prediction intervals (PIs), significance and heterogeneity for each meta-analysis

| Comparison | k | Cohen's d | 95% CI | 95% PI | P | $I^2$ |
|---|---|---|---|---|---|---|
| Pre-ICU scores | 5 | −0.26 | −0.44 to −0.08 | −0.58, 0.07 | 0.005 | 45.50% |
| Community | 10 | −0.22 | −0.43 to 0.00 | −0.88, 0.45 | 0.053 | 87.88% |
| Under 65 | 6 | −0.35 | −0.53 to −0.16 | −0.83, 0.18 | 0.000 | 81.93% |

$I^2$, between study heterogeneity; k, number of independent samples.

### Risk of bias
We found no evidence for publication bias for the community or pre-ICU meta-analyses, from either funnel plots or Egger's test (all p>0.05). Most studies had a moderate degree of methodological quality (13/17). We had insufficient power to explore the effect of study quality on quantitative outcomes.

### Qualitative synthesis
To compare different aspects of QoL, we categorised the subscales into either mental or physical health QoL and calculated a scaled average to enable comparisons (see table 3). Sixteen out of twenty-two studies reported the subscales for both conditions. Our estimates suggest that elderly ICU survivors reported higher average scores on mental health items (mean=57.08/100) than physical health items (mean=47.12/100). Trends in physical health scores compared less favourably to age-matched community controls than did mental health scores (mean differences=−5.23 and −1.71, respectively). Trends in physical health scores were also lower in comparison to younger ICU controls (mean difference=−2.63), whereas mental health scores were higher (mean difference=2.65).

### DISCUSSION
This review has systematically evaluated the literature on QoL for elderly ICU survivors in the medium-to-long term, using EQ-5D composite scores. To our knowledge, this is the first meta-analysis to address this issue. We found evidence that elderly patients who survive ICU can be expected to have slightly worse QoL, compared with younger survivors. To a lesser extent, they may also have worse QoL compared with age-matched community controls and compared with their own QoL up to 1 month before ICU. The wide prediction intervals also suggest that age differences can vary considerably in either direction.

### Strengths in relation to the literature
For the meta-analysis, we identified 2326 elderly ICU survivors within an international sample of 18 cohort studies. We only included recent studies that used validated QoL measures, and we rated most studies as having moderate or higher methodological quality. By pooling these samples using rigorous methods, we have been able to overcome several methodological limitations associated with generalising from individual studies, including small samples, choice of analysis and site selection bias.

Our sensitivity analyses showed that the remaining heterogeneity was partly due to conceptually relevant variables. Given the relatively small literature, these methods ensure that valid, transparent results inform policy and clinical practice decisions.

Although contested, previous reviews have generally concluded that age alone is not a suitable determinant of potential benefit from ICU, especially for survivors.[3 5 20 21] The present study supports these conclusions, although the differences compared with younger ICU survivors (and, to a lesser extent, community samples) are still noteworthy. Decisions on whether to admit patients can be extremely difficult for all involved, with seriously ill elderly people over-represented among the most contentious cases.[22] These challenges are amplified further when healthcare resources are under pressure, such as during the COVID-19 pandemic.

The age-QoL associations we have found may be explained by intermediary variables. Some research suggests that frailty may best explain age differences in QoL following ICU[5 23] and clinical outcome in patients with COVID-19.[24] Frailty is a more integrative approach to conceptualising ageing, but it was not reported within the eligible studies. We would also recommend a meta-analysis of individual patient data for patients with COVID-19, to further stratify clinical variables of interest, including frailty, and to better predict QoL outcomes.

Health economic analysis of ICU in the elderly based on QALYs may be informative when it comes to resource allocation policies, but we have found few such analyses and no explicit polices based on them. They will have to grapple with the controversial notion that everyone is entitled to a 'normal' span of health or 'a fair innings'.[25 26] Given the presumption that a sizeable proportion of elderly survivors will enjoy a good QoL, it is crucial that holistic, person-centred decision-making is not crowded out by survival statistics or anticipatory triage. If triage was to become necessary on the front line, we would advise against weighing age too heavily and rather taking more account of frailty after appropriate consultations.

On average, QoL scores gradually decline with age at approximately 0.5 points per year on the CASP-19 (range 0–57) with a modestly accelerated decrease with older age (>85 years).[4] It is relevant to consider whether change in QoL in the elderly is primarily due to physical health and mental health components. We were unable to incorporate physical and mental subscores into the meta-analysis

**Table 3** An overview of quality of life (QoL) subscores, by mental health and physical health categories, for elderly intensive care unit (ICU) survivors and comparison groups

| First author | Comparison | Measure | Mean MH (elder ICU survivor) | Mean MH (comparison) | Mean difference | Mean PH score (elder ICU survivor) | Mean PH (comparison) | Mean difference |
|---|---|---|---|---|---|---|---|---|
| Anderson | Community | EQ-5D | 58.62 | 55.87 | 2.75 | 47.27 | 48.46 | −1.19 |
| De Rooij | Community | EQ-5D | 56.86 | 58.22 | −1.35 | 48.89 | 50.49 | −1.60 |
| Merlani | Community | SF-36 | 43.00 | 47.00 | −4.00 | 36.00 | 42.00 | −6.00 |
| Jeitziner | Community | SF-36 | 69.72 | 80.37 | −10.65 | 62.71 | 77.91 | −15.20 |
| Villa | Community | SF-36 | 62.40 | 61.50 | 0.90 | 66.60 | 67.90 | −1.30 |
| Garrouste-Orgeas | Community | NHP | 67.13 | 83.00 | −15.87 | 53.63 | 70.23 | −16.60 |
| Schroder | Community | SF-36 | 56.93 | 54.30 | 2.64 | 38.36 | 43.71 | −5.35 |
| Tabah | Community | WHOQOL | 73.30 | 61.40 | 11.90 | 62.10 | 56.70 | 5.40 |
| *Average* | *Community* | | *61.00* | *62.71* | *−1.71* | *51.94* | *57.18* | *−5.23* |
| Grace | Pre-ICU | EQ-5D | 50.80 | 51.40 | −0.60 | 36.30 | 36.90 | −0.60 |
| Cuthbertson | Pre-ICU | SF-36 | 54.00 | 61.67 | −7.67 | 53.22 | 58.50 | −5.28 |
| Hofhuis | Pre-ICU | SF-36 | 51.20 | 50.10 | 1.10 | 38.60 | 38.80 | −0.20 |
| Jeitziner | Pre-ICU | SF-36 | 69.72 | 69.02 | 0.70 | 62.71 | 63.63 | −0.92 |
| *Average* | *Pre-ICU* | | *56.43* | *58.05* | *−1.62* | *47.71* | *49.46* | *−1.75* |
| Abelha | Young | SF-36 | 48.50 | 47.50 | 1.00 | 46.50 | 48.50 | −2.00 |
| Cuthbertson | Young | SF-36 | 51.40 | 51.30 | 0.10 | 37.30 | 37.50 | −0.20 |
| Hofhuis | Young | SF-36 | 51.20 | 50.40 | 0.80 | 38.60 | 38.70 | −0.10 |
| Honselmann | Young | EQ-5D | 51.67 | 51.00 | 0.67 | 44.00 | 54.00 | −10.00 |
| Schroder | Young | SF-36 | 56.93 | 54.30 | 2.64 | 38.36 | 43.71 | −5.35 |
| Eddleston | Young | SF-36 | 63.59 | 58.58 | 5.01 | 58.76 | 63.25 | −4.49 |
| Kleinpell | Young | QLI | 76.26 | 67.93 | 8.32 | 66.33 | 62.60 | 3.73 |
| *Average* | *Young* | | *57.08* | *54.43* | *2.65* | *47.12* | *49.75* | *−2.63* |

All scores were recalculated on a scale of 0–100 (0=minimum QoL; 100=maximum QoL).
MH, mental health; NHP, Nottingham Health Profile; PH, physical health; QLI, Quality of Life Index.

due to differences in the levels of data between measures, so we performed a qualitative synthesis. This suggested that for elderly ICU survivors, mental health questionnaire items were relatively unaffected. The small literature on older adults also suggests relatively low rates of anxiety[27] and depressive disorders,[28 29] although potentially high rates of post-traumatic stress.[30] Further mental health data are needed, as some preliminary reports suggest high rates of post-traumatic stress in ICU patients with COVID-19.[31 32] Our results may serve as a baseline to compare mental and physical health outcomes between COVID-19 and non-COVID-19 survivors.

## Limitations

The primary limitation is the small number of eligible studies for each analysis. To maximise the sample, we included some studies with a small amount of missing data and used validated methods to estimate the mean or the SD from the reported statistics. We argue that these approaches are justified as, based on central limit theorem, we expect the larger sample sizes to produce a better estimate of population variance.[33] For balance, we have also provided a comprehensive overview of our sensitivity analyses to assess risk of bias (see online supplemental appendix). These demonstrate that although our decisions reduced bias, most did not change our interpretation of the effects.

Another potential limitation of the meta-analysis is the focus on long-term ICU survivors, as reported mortality rates were as high as 80% at follow-up. We argue that a substantial 'healthy survivor' effect on QoL is unlikely because survival and QoL have different pathophysiological determinants. We also did not find any evidence of better QoL for elderly patients in studies with high mortality rates. Nevertheless, our results clearly extend only to ICU survivors, rather than prospective ICU patients.

Our results may also be prone to other selection biases. Compared with younger adults, unhealthy elderly adults might be less likely to be admitted into the ICU,[22 34] to survive ICU treatment (possibly in part due to decisions around life-saving treatment)[35] and to survive until follow-up. It was also unclear how many patients had pre-existing cognitive impairments where QoL measurement is more complex, although there was no indication that the proportion was large. Without further data on contextual variables, we would caution wider generalisations to all elderly ICU patients. Nonetheless, these results imply that at least some elderly ICU patients will have a relatively good QoL in the medium-to-long term.

In particular, no patients with COVID-19 were included in the sample. COVID-19 pneumonitis has a specific pathophysiology that does not lead to a 'typical' acute respiratory syndrome, and this can require a relatively high degree of multisystemic involvement. Future studies will need to consider elderly COVID-19 survivors, who often require a relatively lengthy period of ICU treatment and post-ICU rehabilitation, especially if unvaccinated.

We were unable to assess QoL as rigorously as we would have liked. This was partly because studies varied in their definitions of 'old age'. Most of the eligible studies defined this as 65+, following the WHO definition.[36] However, patients aged 65+ currently account for roughly half of all ICU admissions.[37] It is therefore likely that a higher threshold would be more relevant to investigate age-related syndromes. A consensus on what should count as 'very old' would help data collection, analysis and interpretation within this field.

The pre-ICU scores were determined by retrospective ratings from discharged patients or proxies. This is usual practice, but the reliability of proxies is contested.[38 39]

Ideally, we would have analysed differences in QoL change scores between younger and elderly ICU survivors, at multiple time points from before ICU to follow-up.

Finally, we observed moderate-to-high levels of heterogeneity between studies, which limits the generalisability of the results. We found that much of this variation may have been due to mortality rates and length of time post discharge, which supports the view that age alone is not a strong predictor of QoL outcome. We also tried to ensure consistency of measurement by using a mapping function between SF-36 scores and EQ-5D scores, which is a common approach within NICE guidelines.[14 40]

## CONCLUSION

Our study reports the first known meta-analysis of QoL in elderly patients following ICU. We report that on average, elderly survivors of ICU have slightly worse QoL compared with younger ICU survivors, based on physical rather than mental health. To a lesser extent, they may also have worse QoL compared with their own scores before ICU and compared with their community peers. These findings add rigour to the current literature and should inform debates around population-level resource allocation and person-centred intensive care decision-making during the current COVID-19 pandemic and after.

**Author affiliations**
[1]Department of Psychological Medicine, King's College London, London, UK
[2]Critical Care, King's College Hospital NHS Foundation Trust, London, UK
[3]Institute of Mental Health, UCL, London, UK
[4]Dickson Poon School of Law, King's College London, London, UK
[5]Medical Clinic II, Cardiology/Angiology/Intensive Care Medicine, University Hospital Schleswig Holstein, Lübeck, Germany
[6]Department of Emergency Medicine, University Hospital Schleswig Holstein, Lübeck, Germany

**Acknowledgements** We are grateful to Margot Kuylen for her contributions to the reliability assessment and to John Brazier for advising on the SF-36 to EQ-5D mapping function.

**Contributors** KA led at each stage of the project, including drafting the document. KA also acts as guarantor of the study. GO was the primary supervisor on the project, jointly formulated the research questions, led on writing the introduction section and made substantial contributions to all aspects of the study. SC advised on the initial protocol and provided critical revisions from an intensivist perspective. ASD and ARK provided additional supervision and critical revisions. SW also contributed to data collection and analysis, by providing previously unpublished

data, and critical revisions. The manuscript is a transparent account of the study being reported and adheres to PRISMA reporting guidelines. All listed authors have approved for the manuscript to be published in its current format and meet all the ICMJE criteria for authorship. The authors agree to be accountable for the contents of the paper and are jointly responsible for ensuring that all queries related to the accuracy or integrity of the project are investigated and resolved.

**Funding** Supported by the Mental Health and Justice Project, led by GO, which is funded by a grant from the Wellcome Trust (203376/2/16/Z).

**Competing interests** ARK is an adviser on the Faculty of Intensive Care Medicine's Legal and Ethical Policy Unit.

**Patient consent for publication** Not required.

**Provenance and peer review** Not commissioned; externally peer reviewed.

**Data availability statement** Data are available upon reasonable request. All data relevant to the study are included in the article or uploaded as supplementary information. The datasets generated and analysed during the current study are included in this published article and its supplementary information files. Any data queries may also be directed to the corresponding author on reasonable request.

**ORCID iD**
Kevin Ariyo http://orcid.org/0000-0003-0565-2502

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
