## [Reviewer comments · BMJ Open]

ARTICLE DETAILS

TITLE (PROVISIONAL)	Quality of Life in elderly ICU survivors before the COVID-19 pandemic: A Systematic Review and Meta-Analysis of Cohort Studies.
AUTHORS	Ariyo, Kevin; Canestrini, Sergio; David, Anthony; Ruck Keene, Alex; Wolfrum, Sebastian; Owen, Gareth

VERSION 1 – REVIEW

REVIEWER	Flaatten, Hans Haukeland University Hospital, Department of Anesthesia and Intensive Care
REVIEW RETURNED	28-Oct-2020

GENERAL COMMENTS	In this paper the authors have performed a systematic review and meta-analysis of publications dealing with QOL in elderly ICU survivors. They find this group to have reduced QOL compared with younger ICU survivors, but not very different from pre-ICU QOL or QOL in an age-controlled match. Such knowledge is important, since success of intensive care cannot be measured in survival only, but as much in QOL of survivors. This is an important aspect of intensive care in general, and in particular during the present pandemic when the question about who to offer intensive care of the elderly population have been brought forward by many authors. A general comment is the concept of the elderly ICU patient. For some reason this is not discussed in much depth in the present paper, which seems to follow the WHO "definition" of 65 years. Given that median age in most European adult ICUs today is around 70 years, I find this age limit not very relevant for most ICU since it deals with a majority of their patients. Most would not today would consider age 65 years as elderly or old. This is why several research groups have focused to limit studies in patients 75-80 years and above in order to capture the subset of ICU patients where age is commonly associated with many other age related "syndromes". This includes among many frailty, comorbidity, cognitive decline, sarcopenia and reduced ADL. I would for this reason suggest to increase the "bottom line" and include the most important "subset" of patients in this respect. In table 2 it is evident that most included papers in this review have an age limit of 60-65 years (10 studies in table 2). A more specific comment is the sensitivity of their database search. There is at least one important and very large study I miss in this context, and I cannot see why this is not included in this review: Heyland et al. Recovery after critical illness in patients aged 80 years or older: a multi-center prospective observational
---

	cohort study. Intensive Care Med. 2015;41(11):1911-1920. doi:10.1007/s00134-015-4028-2. This study included 610 patients with ICU stay > 24h and followed until one year after ICU discharge. The size of this study would probably have potential to alter the meta-analysis, in particular if studies with patients less old (see above) are removed. I agree with their comment (page 14) that the primary limitation is the small number of eligible studies for each analysis, even if they have “compensated” with including many papers with lower age limit of 60-70 years. If these were excluded the number would half. This finding just underlines a severe deficit in our knowledge about the very old ICU survivors in general, and their quality of life in particular. We also miss data regarding the simultaneous changes in age related “syndromes” along (see above) with quality of life assessment.
--	--

REVIEWER	Cintra, Marco Universidade Federal de Minas Gerais
REVIEW RETURNED	02-Dec-2020

GENERAL COMMENTS	I send some questions: Why did we choose to keep the studies of low quality in the analyzes? Were analyzes performed without low quality studies and verified if the results found change? In the search terms evaluated, keywords associated with frailty were not included, so it may be partly the reason that the theme was not present among the selected studies. I understand that studying frailty was not the aim of the study. However, frail patients tend to have a higher risk of unfavorable evolution in case of severe clinical complications and may progress with a worse outcome and, consequently, a greater chance of functional decline after discharge and a greater risk of worse quality of life. I believe that the question, in addition to being mentioned in the discussion, should be part of the limitations. Most of the selected studies involve elderly patients from European countries. Cultural values can mediate quality of life, so there is a question of generalizing the results for different cultures. Reflecting on the results, it is possible that the “healthy survivor” effect is related to the absence of difference in the pre-morbid quality of life and after admission to the ICU. Although survival and quality of life have different pathophysiology, considering whether unevaluated variables, such as the degree of autonomy and independence of the elderly after discharge from the ICU, the “healthy survivor” effect may have indirectly affected the results. Finally, the need for studies with a larger sample and different cultures and with an appropriate methodology on the topic should be described.
--

REVIEWER	Pripp, Are Hugo Oslo universitetssykehus Ulleval, Oslo Centre for Biostatistics & Epidemiology
REVIEW RETURNED	09-Feb-2021

GENERAL COMMENTS	I have mainly assess the statistical analysis and methodology and found them appropriate and well performed. As the authors point out, the small number of studies for each analysis is a limitation.
---

	However, this is often the case in many meta-analysis. They conducted several sensitivity analysis. Minor comment: Please indicate in the forest plot the direction / interpretation of the effect estimate. For example, in Figure A.1. is a negative effect estimate a better or lower life quality at post-ICU compared to pre-ICU. I assume it is a lower life quality at post-ICU compared to pre, but some readers may only look at the Forest plot and they should be as self-explanatory as possible. Further, I suggest to add some heterogeneity statistics to each Forest plot figure, e.g. I2 statistics.
--	---

VERSION 1 – AUTHOR RESPONSE

N	Reviewer	Reviewer Comment	Initial Comments
1.	Associate Editor	We don't know the indications for ICU admission in the individual studies.	All of our included studies used mixed pathology samples. We had already broken down the type of unit (page 7, lines 13-15). We have included a breakdown of the reasons for admission in the appendix, and cited this in page 7, line 15. Note that the manuscript generally reports the breakdown at study level, ie. including non-survivors and people who did not complete QoL measures at follow up.
2.	Associate Editor	For instance, this doesn't include any studies of covid patients, but not sure we will see similar findings in covid patients, who will often require a prolonged period of rehabilitation following ICU admission and whose quality of life will necessarily be affected.	See point 4 for our suggested sentence.
3.	Associate Editor	Quality of life may also differ depending on the duration of ICU admission (it's very different being admitted for a few days compared to several months like is the case with some covid patients).	We did test whether variation in average length of stay, in either ICU or the hospital, affected the results. This was non-significant with the available data. Note that this analysis had moderate missing data, and LoS was mostly measured at the study level. However, we do acknowledge that the average LoS in our samples was much lower than the average COVID sample. See point 4 for our suggested revision.
4.	Associate Editor	Can you comment on that? These should at least be acknowledged as limitations.	We have suggested the revisions below Page 2, line NA:

			“Our sample had a mixture of conditions, and because data was reported inconsistently and often at study-level, it was difficult to generalise to specific clinical groups, including COVID-19 patients” Page 13, lines 40-44: “Data describing QoL at follow-up of elderly survivors admitted to ICU with a diagnosis of COVID-19 were not available at the time of data extraction, we were therefore unable to include in the sample this sub-group of patients. Future studies will need to consider elderly COVID survivors, who often require a relatively lengthy period of ICU treatment and post-ICU rehabilitation.”
5.	Editor	Please ensure that your abstract is formatted according to our Instructions for Authors:	Done. We have revised headings as advised.
6.	Editor	Please revise the ‘Strengths and limitations’ section of your manuscript (after the abstract). This section should contain five short bullet points, no longer than one sentence each, that relate specifically to the methods. The results of the study should not be summarised here.	Done. We have revised to five shorter bullet points focusing on the methods, as advised.
7.	Editor	Please re-upload your Supplementary files in PDF format.	Done.
8.	Editor	Please remove all your figures in your main document and upload each of them separately under file designation ‘Image’ (except tables and please ensure that Figures are of better quality or not pix-elated when zoom in). NOTE: They can be in TIFF, JPG or PDF format and make sure that they have a resolution of at least 300 dpi. Figures in DOCUMENT, EXCEL and POWERPOINT format are not acceptable.	Done.
9.	Editor	The in-text citation for “Figure 1” is missing in the main text of your main document file. Please amend accordingly.	Done.
10	Editor	We have implemented an additional requirement to all articles to include ‘Patient and Public Involvement’ statement within the main text of your main document.	Done.

		Please refer below for more information regarding this new instruction	
11	Reviewer One	In this paper the authors have performed a systematic review and meta-analysis of publications dealing with QOL in elderly ICU survivors. They find this group to have reduced QOL compared with younger ICU survivors, but not very different from pre-ICU QOL or QOL in an age-controlled match. Such knowledge is important, since success of intensive care cannot be measured in survival only, but as much in QOL of survivors. This is an important aspect of intensive care in general, and in particular during the present pandemic when the question about who to offer intensive care of the elderly population have been brought forward by many authors.	N/A
12	Reviewer One	A general comment is the concept of the elderly ICU patient. For some reason this is not discussed in much depth in the present paper, which seems to follow the WHO "definition" of 65 years. Given that median age in most European adult ICUs today is around 70 years, I find this age limit not very relevant for most ICU since it deals with a majority of their patients. Most would not today would consider age 65 years as elderly or old. This is why several research groups have focused to limit studies in patients 75-80 years and above in order to capture the subset of ICU patients where age is commonly associated with many other age related "syndromes". This includes among many frailty, comorbidity, cognitive decline, sarcopenia and reduced ADL. I would for this reason suggest to increase the "bottom line" and include the most important "subset" of patients in this respect. In table 2 it is evident that most included papers in this review have an age limit of 60-65 years (10 studies in table 2).	Dear Prof Flaatten, thank you for your comments. The point you make, one we discussed in length during the planning, is obviously of fundamental importance and reflected by the current trend in the literature. We agree that categorising as "old" patients over 65 is problematic. It gives the impression that we are dealing with "a" subgroup, while the over 65 account for a significant proportion, where not the majority, of ICU patients. Also, the idea that, at least in the Western society, a 10-year gap between, for instance, 56 and 66 should be considered pathophysiologically meaningful is questionable. That is why, as you suggest, because "a" threshold will have to be set, 75 or 80 would single out those whose drop in physiological reserve is (at the sub-population level) more relevant (in comparison with their younger counterpart) for the transformations expected at that stage of life. So we agree that the shift observed recently towards the analysis of those defined in the literature as the "very old" (a threshold variably set at 75, 80 and 85) should be encouraged. Our reason for adopting the threshold of 65 (even 60 for one study), in the face of the above observations is, as you have mentioned, contingent on the available literature that, until recently, has considered old those older than 65.

			To mitigate this issue we will: (1) contact study authors who may have unpublished eligible data for “very old” patients (including the Heyland study; see point 13) (2) suggest in the conclusion that a higher threshold would be more relevant (see page 16, lines 46-50) (3) that a consensus on what should count as very old (whether 75, 80 or 85) would help collection, analysis and interpretation of data. (see pages 12-13, lines 49-2); and (4) expanded the moderator analysis for ‘minimum age’ to include the 75-80 subgroup in the community control analysis (see page 9, lines 2-7). This was not possible for other analyses, as the control samples were too varied. We also note that we conducted a moderator analysis in the previous analysis, to investigate whether the effect varied on the basis of the minimum age of the sample, which was not statistically significant.
13	Reviewer One	A more specific comment is the sensitivity of their database search. There is at least one important and very large study I miss in this context, and I cannot see why this is not included in this review: Heyland et al. Recovery after critical illness in patients aged 80 years or older: a multi-center prospective observational cohort study. Intensive Care Med. 2015;41(11):1911-1920. doi:10.1007/s00134-015-4028-2. This study included 610 patients with ICU stay > 24h and followed until one year after ICU discharge. The size of this study would probably have potential to alter the meta-analysis, in particular if studies with patients less old (see above) are removed.	Thank you, we note your concern that potentially relevant studies could not be included in the meta-analysis. Unfortunately, we excluded nine studies (including Heyland et al) because, although they studied eligible participants, they did not report the figures for QoL in the manuscript. We have contacted each of these authors to see whether we could add any unpublished data to the meta-analysis. One study author responded with their dataset, which we have included (see Honselmann et al). Other authors of these papers (eg. Prof. Chelluri, Dr. Wildman, Prof. Carlet) responded to say that they could not locate the data. The paper has been revised in light of these updated results:  • Abstract

			 • Figure 1 • Page 4 lines 24-25 • The results and discussions sections
14	Reviewer One	I agree with their comment (page 14) that the primary limitation is the small number of eligible studies for each analysis, even if they have “compensated” with including many papers with lower age limit of 60-70 years. If these were excluded the number would half. This finding just underlines a severe deficit in our knowledge about the very old ICU survivors in general, and their quality of life in particular. We also miss data regarding the simultaneous changes in age related “syndromes” along (see above) with quality of life assessment.	We agree with the reviewer that the current research literature on QoL is limited. As you mentioned, our decisions reflect our desire to have statistical power, in addition to a valid sample of older adults. We hope that, through recommending better reporting and a standard definition what constitutes ‘old/very old’ patients in ICUs (see point 12), we can contribute to reducing this deficit in knowledge.
15	Reviewer Two	Congratulations on the study, I believe it is a relevant question and not properly evaluated in the literature.	N/A
16	Reviewer Two	Why did we choose to keep the studies of low quality in the analyzes? Were analyzes performed without low quality studies and verified if the results found change?	Thank you for raising this. The main reason was because any moderator analysis would have been severely underpowered. We came to this conclusion due to lack of variation within the sample. The sample sizes were small for each analysis (n= 5-9), and most of the studies (70%) were of medium quality. Furthermore, the quality measures are mostly used for categorical outcomes (high/medium/low) rather than continuous outcomes. These factors would have led to a high probability of type one or type two errors, which would have made any outcome difficult to interpret. Therefore, we didn’t consider a quantitative analysis of study quality to have been particularly meaningful. We decided it would have been more transparent to present the quality assessment for each study.
17	Reviewer Two	In the search terms evaluated, keywords associated with frailty were not included, so it may be partly the reason that the theme was not present among the selected studies.	As the reviewer mentions, Age and QoL were our primary variables of interest, and we consider frailty to be one of many possible contextual variables (see point 18). Due to this, we did not include frailty within our search terms. We are confident that, between the original search, and the forward citation and reference list

			searches, we have included all studies that mention frailty while also meeting our other eligibility criteria. Unfortunately, studies that did not meet our eligibility criteria (eg. that did not report statistics on QoL) are unfortunately outside the scope of the meta-analysis, as they would not address our primary research question.
18	Reviewer Two	I understand that studying frailty was not the aim of the study. However, frail patients tend to have a higher risk of unfavorable evolution in case of severe clinical complications and may progress with a worse outcome and, consequently, a greater chance of functional decline after discharge and a greater risk of worse quality of life. I believe that the question, in addition to being mentioned in the discussion, should be part of the limitations.	We agree that frailty is indeed very relevant. As the reviewer mentions, age and QoL were our primary variables of interest, and we mention our inability to address various contextual variables within the discussion. We explicitly refer to frailty at length, in page 11, lines 32-37. However, we would argue that multiple other variables would be at least as relevant as frailty (e.g. severe COPD, IHD and so on). Patients are extremely different in a multitude of characteristics, which are all potentially relevant in respect to QoL. These could not be studied as they were not reported consistently. We therefore also refer the author to page 13, lines 35-36, in which we make a more inclusive point, that the lack of contextual variables is a limitation.
19	Reviewer Two	Most of the selected studies involve elderly patients from European countries. Cultural values can mediate quality of life, so there is a question of generalizing the results for different cultures.	Thank you for pointing out this important aspect. Cross country cultural variations are certainly relevant in the interpretation of the results. However, for QoL is not measurable in itself we are forced to employ surrogates in its place (a number of measurable dimensions). It is assumed, we agree, that they would then have to be interpreted through the lens of every culture (every person for that matter) in the most appropriate way (in other words any “culture” would have to judge how much QoL stems from those measured/measurable dimensions). In addition, we would argue that, if our study were to

			be replicated in another “culture”, the same dimensions (scores) would have to necessarily be employed; in other words, from a methodological prospective, there would not be currently a strategy to embed the “cultural” dimension into the methodology. It is also important to note that our effect measures are difference scores, rather than raw scores. In other words, each sample of elderly survivors is ‘controlled for’ by a group within the same local area, or indeed the same person. This should render the results more generalisable. Finally, both the highest and lowest EQ-5D VAS values have been found in European countries (in Hungary and Denmark, respectively; link here). This runs counter to the assumption that a non-European sample would yield very different raw scores. We would therefore consider this reflection very relevant, but not necessarily a limitation, because the EQ-5D is valid for use across cultures.
20	Reviewer Two	Reflecting on the results, it is possible that the “healthy survivor” effect is related to the absence of difference in the pre-morbid quality of life and after admission to the ICU. Although survival and quality of life have different pathophysiology, considering whether unevaluated variables, such as the degree of autonomy and independence of the elderly after discharge from the ICU, the “healthy survivor” effect may have indirectly affected the results.	Thank you for pointing this out. We agree that there is potential for selection bias, as previously noted in the limitations section (currently page 12, lines 31-39). We believe that our point on ‘contextual variables’ is analogous to ‘unevaluated variables’ and we have therefore acknowledged your point.
21	Reviewer Three	I have mainly assess the statistical analysis and methodology and found them appropriate and well performed. As the authors point out, the small number of studies for each analysis is a limitation. However, this is often the case in many meta-analysis. They conducted several sensitivity analysis.	N/A
22	Reviewer Three	Please indicate in the forest plot the direction / interpretation of the effect estimate. For example, in Figure A.1. is a negative effect estimate a better or	Done. See relevant figures.

lower life quality at post-ICU compared to pre-ICU. I assume it is a lower life quality at post-ICU compared to pre, but some readers may only look at the Forest plot and they should be as self-explanatory as possible. Further, I suggest to add some heterogeneity statistics to each Forest plot figure, e.g. I2 statistics.

VERSION 2 – REVIEW

REVIEWER	Flaatten, Hans Haukeland University Hospital, Department of Anesthesia and Intensive Care
REVIEW RETURNED	13-Apr-2021

GENERAL COMMENTS	Thank you for responding to my comment in the first version, and I find your answers to my satisfaction. I ave just thre minor comments: 1. Page 6, line 11. Please explain M, is not intuitive to me Median?? this also occurs on page 9
--

REVIEWER	Cintra, Marco Universidade Federal de Minas Gerais
REVIEW RETURNED	20-Apr-2021

GENERAL COMMENTS	I forward my recommendations:  1. I suggest including in the design of the abstract that this is a meta-analysis; 2. I believe that the introduction should be more incisive on the need to define the criteria not related to the age of who will be admitted to an ICU in situations of health system collapse, as is happening in the pandemic by COVID-19, the so-called "Sophie's Choice "; 3. Why were studies carried out in countries that are not OECD members excluded? 3. In the methodology, it should be mentioned that the search terms used are described in item 6 of the appendix. 4. Cite in the methodology that the inclusion and exclusion criteria are detailed in item 5 of the appendix. 5. Table 1 needs corrections. The Hofhuis 2011 and Jeitziner studies are duplicated, that is, described twice. The article cites a sample of 2585 elderly people, but disregarding only duplicate studies, the sum of the studies in the table results in 2679 people. I recommend that the entire table and analysis be reviewed. 6. As for the discussion, there is no doubt that quality of life should be used as a criterion for admission to the intensive care sector. But I understand that the results do not support the use of quality of life as a single tool for prioritizing elderly people with indication for intensive care. It should be a tool to be associated with the assessment of clinical severity and frailty.
--

VERSION 2 – AUTHOR RESPONSE

N	Reviewer	Reviewer Comment	Response
1.	1	Page 6, line 11. Please explain M, is not intuitive to me Median?? this also occurs on page 9.	M is the mean. We have spelled this out on page 9
2.	1	I suggest including in the design of the abstract that this is a meta-analysis;	Done. See abstract for edits.
3.	2	I believe that the introduction should be more incisive on the need to define the criteria not related to the age of who will be admitted to an ICU in situations of health system collapse, as is happening in the pandemic by COVID-19, the so-called "Sophie's Choice ";	We must note that non-age variables were not part of the main research questions, and therefore, we do not directly address them in the study. We would also reiterate that in the section outlining the research questions, on page 3 lines 29-41, we acknowledged that we sought to model variables that are not related to age. For these reasons, we have only briefly expanded on our explanation of these issues, to include a) a paper on the ethical debate on resource allocation and b) the context of difficult admission decisions during COVID-19. Kuylen, M. N., Kim, S. Y., Keene, A. R., & Owen, G. S. (2021). Should age matter in COVID-19 triage? A deliberative study. Journal of Medical Ethics, 47(5), 291-295.
4.	2	Why were studies carried out in countries that are not OECD members excluded?	We excluded non-OECD countries to minimise risk of bias. OECD countries make up the vast majority of all studies on intensive care. They also have similar characteristics on key variables that are relevant to the present study, such as life expectancy and healthcare spending. There was no indication that this was a large literature, and so we wanted to ensure comparability by excluding non-OECD countries. It is common practice for systematic reviews of healthcare interventions to limit selection based on socioeconomic criteria, such as OECD membership. See, for example: Olanrewaju, O., Kelly, S., Cowan, A., Brayne, C., & Lafortune, L. (2016). Physical activity in community dwelling older people: a systematic review of reviews of interventions and context. PloS one, 11(12), e0168614

			Sauvé-Schenk, K., Duong, P., Savard, J., & Durand, F. (2020). A systematic review of social service and community resource interventions following stroke. Disability and Rehabilitation, 1-10. Finally, this helped to ensure our results were comparable with similar reviews that had only OECD samples, including: Oeyen, S. G., Vandijck, D. M., Benoit, D. D., Annemans, L., & Decruyenaere, J. M. (2010). Quality of life after intensive care: a systematic review of the literature. Critical care medicine, 38(12), 2386-2400.
5.	2	In the methodology, it should be mentioned that the search terms used are described in item 6 of the appendix.	Done. See page 4, lines 9-10 for edits.
6.	2	Cite in the methodology that the inclusion and exclusion criteria are detailed in item 5 of the appendix.	Done. See page 4, lines 15-16 for edits.
7.	2	Table 1 needs corrections. The Hofhuis 2011 and Jeitziner studies are duplicated, that is, described twice. The article cites a sample of 2585 elderly people, but disregarding only duplicate studies, the sum of the studies in the table results in 2679 people. I recommend that the entire table and analysis be reviewed.	We have reviewed each of the tables. The details in Table 1 are correct, barring from one minor edit to the sample size. We re-ran the analysis, but this made no difference to results. However, we noticed that one study had been included in error in the meta-analysis, rather than the qualitative analysis (Cuthbertson, 2010). This was because the mapping algorithm only applies to individual scores, rather than the MCS or PCS component scores reported by this study. We have revised the results and manuscript accordingly. This has only changed one analysis, which became statistically significant, but made no real difference to the effect size. It therefore makes minimal difference to the interpretation of the results and remains in the qualitative synthesis. We have made minor corrections to two figures in Table 3 (one as a result of the above).

			The Hofhuis and Jeitziner studies were reported accurately. They are described twice because those studies each reported two different outcomes included in the analysis. However, we acknowledge that the previous format may have been confusing, and so we have made this clearer in the table. This includes separating out the two samples in the Honselmann study. We cite the new sample size on page 6, line 10. The correct sum is now 2326, because the Cuthbertson study is now excluded from the meta-analysis.
8.		As for the discussion, there is no doubt that quality of life should be used as a criterion for admission to the intensive care sector. But I understand that the results do not support the use of quality of life as a single tool for prioritizing elderly people with indication for intensive care. It should be a tool to be associated with the assessment of clinical severity and frailty.	We fully agree with this comment. Clinical severity and frailty predict survival and these are both relevant to admission criteria. The results do not support the use of age alone as admission criteria. We regard this as consistent with our discussion.

VERSION 3 – REVIEW

REVIEWER	Cintra, Marco Universidade Federal de Minas Gerais
REVIEW RETURNED	18-Jul-2021
GENERAL COMMENTS	My position is to accept the article.